# Synergistic Action of *Cinnamomum verum* Essential Oil with Sertraline

**DOI:** 10.3390/antibiotics11111617

**Published:** 2022-11-13

**Authors:** Alexia Barbarossa, Sabina Sblano, Antonio Rosato, Antonio Carrieri, Filomena Corbo, Maria Lisa Clodoveo, Giuseppe Fracchiolla, Alessia Carocci

**Affiliations:** 1Department of Pharmacy—Drug Sciences, University of Bari “Aldo Moro”, 70125 Bari, Italy; 2Interdisciplinary Department of Medicine, School of Medicine, University of Bari “Aldo Moro”, 70124 Bari, Italy

**Keywords:** essential oil, antimicrobials, synergism, *Cinnamomum verum*, checkerboard microdilution method, sertraline, drug repositioning

## Abstract

*Cinnamomum verum* L. essential oil (CEO), commonly known as Ceylon cinnamon or cinnamon tree, is regarded as one of the most employed essential oils in the field of aromatherapy. It is usually applied externally as astringent, antipruritic, rubefacient, and anti-septic agent. Furthermore, both in vitro and in vivo research have demonstrated its numerous pharmacological effects, including the potentiality for treating neuralgia, myalgia, headache, and migraine. Several pieces of research also corroborated its significant antiviral and antimicrobial properties. Cinnamaldehyde, eugenol, caryophyllene, cinnamyl acetate, and cinnamic acid are the most representative compounds that are generally found in greater quantities in CEO and play a pivotal role in determining its pharmacological activities. Due to the global antibiotic resistance scenario and the dwindling amount of funding dedicated to developing new antibiotics, in recent years research has concentrated on exploring specific economic approaches against microbial infections. In this context, the purpose of this study was the investigation of the synergistic antibacterial activities of commercially available and chemically characterized CEO in combination with sertraline, a selective serotonin reuptake inhibitor (SSRI), whose repositioning as a non-antibiotic drug has been explored over the years with encouraging results. In vitro effects of the titled combination were assessed toward a wide panel of both Gram-positive and Gram-negative bacteria. The antimicrobial efficacy was investigated by using the checkerboard microdilution method. The interesting preliminary results obtained suggested a synergistic effect (fractional inhibitory index, FICI < 0.5) of sertraline in combination with CEO, leading to severe growth inhibition for all bacterial species under investigation.

## 1. Introduction

The emergence and spread of drug-resistant pathogens, resulting in antimicrobial resistance, continue to compromise our capability of handling commonly occurring infectious diseases. Of particular concern is the accelerated worldwide diffusion of multidrug-resistant bacteria that provoke infections that are not treatable with existing antimicrobial drugs, such as antibiotics [1,2]. Drug-resistant bacteria are a serious global public health challenge, requiring new antimicrobial agents on an urgent basis. In recent years, many research programs have concentrated on the development of novel compounds with possible antimicrobial effects to address this issue [3,4,5,6,7], and new sources, such as plant-derived antimicrobial compounds, have been thoroughly investigated [8,9,10,11,12,13]. Essential oils (EOs) have been extensively studied for their pharmacological properties, since they could be a promising source of new natural remedies. EOs exhibit antifungal, antibacterial, and antiviral activities and have been examined on a global scale as the possible origin of novel antimicrobial compounds, agents promoting food preservation, and different options for handling infections [14,15,16,17,18,19]. Studies in the literature have widely reported the mechanism of action of EOs that relates to the breakdown of the bacterial cell wall, although the impact on the demolition of enzymes or membrane proteins or overflow of cell contents after cytoplasmic membrane failure is also plausible [20]. Based on this evidence, synergistic effects between antibiotics and EOs against bacteria can take place. This element is significant because it will implicate the reduction in the use and the dosage of antibiotics in therapies, reducing their side effects. Furthermore, exploiting the synergism between EOs and antibiotics could represent an answer to overcome antimicrobial resistance [21,22]. Currently, about 3000 essential oils are known, 300 of which are available on the market in the fields of agriculture, food, health, and pharmacy [23,24,25]. Plant-based EO is a complex natural mixture that may contain over 100 components at entirely different amounts. These components comprise two groups of diverse biosynthetic sources: the principal group consists of terpenes and terpenoids, and the other group consists of low-molecular-weight aromatic and aliphatic components [26,27]. Although major components are generally responsible for EOs biological activities, the contribution of minor constituents to these activities should also be considered. Among the various essential oils of botanical origin that have potential antimicrobial activity include those obtained from the species of the genus *Cinnamomum* (Lauracee), such as *Cinnamomum verum*, commonly known by the popular and commercial name of Ceylon cinnamon or cinnamon tree due to the most famous country origin of the cinnamon bark spices [28]. Ceylon cinnamon barks are used in traditional medicine to relieve asthma, bronchitis, diarrhoea, headache, inflammation, and cardiac disorders. Various parts of the plant, such as the bark, leaves, and flowers, are employed to produce EO by steam distillation process. Cinnamaldehyde, eugenol, caryophyllene, cinnamyl acetate, and cinnamic acid are the main compounds found in the CEO. The pharmacological activities of these compounds are very varied. The antiviral, antibacterial, and antioxidant properties of CEO have been shown [29,30,31]. Moreover, several studies investigated the use of EOs to boost the efficacy of antibiotics as a different approach to deal with infections, resulting from drug-resistant bacteria [32]. Studies by Yap et al. indicated the antimicrobial activity of *C. verum* bark EO, both alone and associated with piperacillin, against a multidrug-resistant *Escherichia coli* strain [33].

In our previous studies we showed the synergistic effect of several EOs in combination with some commercially available antimicrobials, demonstrating the efficacy of these combinations and suggesting the potential of a new therapeutic use [34,35,36,37]. Recently, different medicinal products of disparate therapeutic classes are being studied as antimicrobials in the drugs-repurposing approach that is an interesting alternative to treating infectious diseases provoked by multidrug-resistant pathogens. These compounds have been named “non-antibiotic drugs” [38]. The antimicrobial effects of different non-antibiotic drugs have been reported, including statins and non-steroidal anti-inflammatory drugs (NSAIDs), as well as cytostatics and psychotropics [39,40]. Some antidepressant drugs demonstrated antimicrobial effects, being the highest activity evidenced in the third-generation antidepressants known as selective serotonin reuptake inhibitors (SSRIs), such as fluoxetine, paroxetine, and sertraline [41]. Among them, sertraline, due to its antimicrobial potential, enhanced the activity of different antibiotics, counteracted the multidrug resistance of pathogens, and made them susceptible to previously resistant drugs [42,43,44,45]. The association of repositioned drugs with EOs is also a valuable method for a prompt recognition of novel therapies to treat acute infections. In a recent study, we demonstrated the synergistic antifungal effect of diclofenac, a widely utilized NSAID, in combination with EOs endowed with antimicrobial activity towards numerous strains of *Candida* spp. [46]. This evidence prompted us to evaluate the potential antibacterial synergistic effect of a new combination of sertraline and CEO, with the aim of providing greater effectiveness in the fight against infections and overcoming drug resistance.

## 2. Results

### 2.1. Cinnamon verum EO Chemical Composition

The chemical composition of CEO has been determined by GC/MS analysis [47]. The analysis resulted in the identification of 99% of the whole mixture that comprises 18 chemical compounds reported in Table 1. The major constituents of CEO are cinnamaldehyde (73%), linalool (6.78%), eugenol (6%), cinnamyl acetate (5%), and eucalyptol (1.47%). Other constituents are present in less than 1% i.e., p-cymene, humulene, *α*-pinene, etc. Our results agree with other studies that reported cinnamaldehyde as the major chemical compound of CEO [48,49].

### 2.2. Antibacterial Activity

In this study, we investigated the synergistic antibacterial activity of CEO in combination with the sertraline (SSRI) against several strains of Gram-positive and Gram-negative bacteria. All the minimal inhibitory concentration (MIC) values for sertraline and CEO are reported in Table 2. MICo is the MIC value of a single component tested alone, while MICc is the MIC value of each component in the association at the most effective inhibition growth. Fractional inhibitory concentration (FIC) is determined by the ratio MICc / MICo and the fractional inhibitory concentration index (FICI) is determined by adding together the FIC of sertraline and FIC of CEO. In the last column of Table 2, R% represents the percentage reduction in the amount of each associated component, compared to each single component [55].

FICI values for the association were in the range of 0.08–0.43, indicating a clear-cut synergism between CEO and sertraline against all tested bacterial strains. It is worth noting that the MIC value (MICc) for sertraline is significantly decreased when combined with EO, since the MIC value for this association was found to be more than 30-fold lower than most of the bacterial species considered. Among the Gram-positive bacteria, the most interesting results have been obtained for *Enterococcus faecalis* ATCC 29212 and *Enterococcus faecalis* bs for which the MIC value of sertraline was found to decrease from 16.0 (MICo) to 0.5 (MICc) μg/mL (FICI = 0.08) and from 32.0 (MICo) to 1 (MICc) μg/mL (FICI = 0.08), respectively. Furthermore, a promising result was also obtained against *Acinetobacter baumannii* ATCC 19606, a Gram-negative bacillus resistant to treatment with different antibiotics. In this case, a 32-fold reduction in sertraline MIC (FICI = 0.08) was observed when used in combination with EO too. The strong synergy observed between sertraline and the EO against the Gram-negative *Pseudomonas aeruginosa* ATCC 27853 (FICI = 0.11) and *Klebsiella pneumoniae* ATCC 19833 (FICI = 0.43) is worthy of note. In particular, the MICc value for sertraline is much lower than what is normally required to achieve the direct inhibition of bacterial growth (MICc: 64 μg/mL vs. MICo: 8μg/mL and MICc: 33 µg/mL vs. MICo: 1.03 µg/mL, respectively). Sertraline combinations with EO also exhibited a pronounced synergistic effect against *Escherichia coli* ATCC 25922 (FICI = 0.23) and *Bacillus subtilis* ATCC 6633 (FICI = 0.11). Another way to analyze the results is to evaluate the reduction in the amount of compounds used in the association; this important aspect is evident for all the species, for example, the table shows that a 16-fold reduction in sertraline for *Bacillus subtilis* ATCC 6633 (FICI = 0.11) and a 32-fold reduction for *Staphylococcus aureus* ATCC 43300 (methicillin-resistant *Staphylococcus aureus*) (FICI = 0.43) have been observed. The combination of sertraline and EO shows a marked synergistic effect against *Escherichia coli* ATCC 25922 (FICI = 0.23) and *Corynebacterium striatum* RM (FICI = 0.08). The synergistic interactions between CEO and the promising antimicrobial, sertraline, in inhibiting eight out of the fifteen bacterial strains under study are reported in Figure 1 and Figure 2 by the isobologram [56]. The isobole curves appear as concave lines, with the additive and synergic effects of the combined agents against the bacterial strains being considered [57].

## 3. Discussion

The use and discovery of antibiotics represent a milestone in medicine, but the onset of drug resistance to antibiotics is becoming a serious emergency that needs to be dealt with. For this reason, research is turning its attention to the approach of repositioning drugs already known for other therapeutic properties and which may show off-label effects, such as potential antibiotics. An effective combination of therapy to treat bacterial infectious diseases may reach wider antibacterial coverage and potentially reduce acquired resistance. The combination of repositioned medications and EOs is also an interesting approach to rapidly identify new treatments for acute infections. Several studies have shown that the antidepressant sertraline, besides its intended use, demonstrates an antimicrobial effect against several bacteria strains alone or in combination with antibacterial agents. The exact mechanism by which this occurs is still not clear, conceivably, as sertraline is a selective serotonin reuptake inhibitor, a reuptake pump inhibitor in humans, so it can act as an efflux pump inhibitor in bacteria [42,45]. 

Because of their potential therapeutic effects, EOs are widely used as alternative antimicrobial agents for a variety of infections. Our previous EOs studies demonstrated their synergy with some commercially available antibiotics and with diclofenac, a well-known non-steroidal anti-inflammatory drug, which exhibited a good antimicrobial and antifungal activity. Our studies proved the efficacy of these associations by supporting the possibility of a new therapeutic use [35,36,37,46]. In this study, we reported the effect of sertraline in combination with CEO on several bacterial strains, the antimicrobial activity of which has been extensively demonstrated. As underlined in our in vitro assays, the bacterial strains under study displayed their sensitivity to the compounds tested, both individually and in combination, as can be seen in Table 2. The findings confirm the synergy between sertraline and CEO. Although it is not possible to establish the occurrence of adverse effects in this study as it is a preliminary in vitro study, the reduction in the doses of both components of the combination and the different mechanisms of action with which the antimicrobial activity of sertraline would develop is a promising result for the limitation of the side effects of sertraline itself as an antidepressant.

Indeed, there is clear evidence of a significant reduction in the active sertraline concentration when used in combination with EO for all bacterial strains tested. It should be noted that the MIC for sertraline is reduced from 16.0 µg/mL to 0.5 µg/mL for *E*. *faecalis* ATCC 29212, *S. aureus* ATCC 25923, and *S. aureus* ATCC 29213. Similar results have been obtained for the Gram-negative bacteria *A. baumannii* ATCC 19606 and *E. coli* ATCC 25922. The active sertraline concentration was also significantly reduced for all other bacterial strains tested. Conceivably, this effect should be ascribed to the presence of fundamental active compounds in EO acting in association with sertraline. The elucidation of the exact mechanism by which the observed synergism between sertraline and CEO occurs will require molecular studies. The mechanism of action is conceivably multifactorial, deriving from the presence of fundamental active compounds in EO acting in association with sertraline. As reported in the literature, the synergy of EOs could be explained by their ability to disrupt the permeability barrier of the microbial plasma membrane [32,58]. This disruption may promote the entry of sertraline into the microbial cell, thus, interacting with the bacterial target and ultimately generating its antibacterial effect.

## 4. Materials and Methods

The chemical composition of commercially available CEO used in our experiments was confirmed by GC/MS analyses [47]. The antibacterial activity of CEO and sertraline against several Gram-positive and Gram-negative bacteria, along with their synergistic effects, have been studied by following the microdilution checkerboard method. 

### 4.1. Materials

CEO (Lot 140/0000324, 10.2018, 10 mL) was provided by Erbe Nobili srl (Corato, Bari, Italy) and was stored in a vial protected from light and at 0–4 °C until use. Sertraline, all the analytical standard (Table 1), C7–C30 alkanes mixture, and all the solvents (HPLC grade) were purchased from Sigma-Aldrich srl (Milan, Italy). Filters used were supplied by Agilent Technologies Italia spa (Milan, Italy). The culture media used are Mueller Hinton Broth (Oxoid S.p.A., Rodano, Milano, Italy) and Mueller Hinton Agar (Oxoid S.p.A., Rodano, Milano Italy).

Fifteen bacterial strains from American Type Culture Collection (ATCC, Rockville, MD, USA) and clinical isolates were used for tests: *Bacillus subtilis* ATCC 6633, *Enterococcus faecalis* ATCC 29212, *Staphylococcus aureus* 25923, *Staphylococcus aureus* ATCC 29213, *Staphylococcus aureus* ATCC 43300 (MRSA), *Staphylococcus aureus* ATCC 6538p, *Acinetobacter baumannii* ATCC 19606, *Escherichia coli* ATCC 25922, *Klebsiella pneumoniae* ATCC 13883, *Pseudomonas aeruginosa* ATCC 27853, *Escherichia coli* ESBL, *Corynebacterium striatum* RM, *Enterococcus faecalis* BN21, *Staphylococcus aureus* BS, *Enterococcus faecalis* BS.

All the clinical isolates were from patients admitted to the intensive care unit of the Department of Biomedical Sciences and Human Oncology, University of Bari “Aldo Moro”, Italy. The isolation procedures were conducted in the Hygiene Section of the Department, using conventional physiological and morphological methods and the API system.

### 4.2. Methods

#### 4.2.1. Antimicrobial Activity

The bacterial species were cultured on Mueller Hinton Agar (MHA), and each bacterial suspension was composed of 23 colonies for each strain taken from an MHA plate and dissolved in 2 mL of Mueller Hinton Broth (MHB ). The suspensions obtained were diluted with 0.85% NaCl solution, then adjusted to 1 × 10^8^ CFU/mL (0.5 McFarland). The bacteria were subjected to two sub-cultures of MHA prior to testing. Each bacterial suspension was taken from its frozen stock at −70 °C. The strains were inoculated in 5 mL of Muller Hinton broth and then incubated under stirring at 35 °C for 48 h. MIC values were determined by broth microdilution method, in accordance with CLSI (Clinical and Laboratory Standards Institute) Protocol M07A9 guidelines [59,60]. For the antibacterial test, a stock solution (EO/Ethanol 1:2.5, 40% *v*/*v* with Tween 80, 0.1%) was diluted 1:20 in MHB to obtain a 2% (*v*/*v*) final solution. Doubling dilutions of the EO from 2% to 0.015% for EO were prepared directly in 96-well microtiter trays in MHB. After the addition of 0.1 mL of the inoculum, the microtiter trays were incubated at 36 °C for 24 h. The final concentration of ethanol was 1.5% (*v*/*v*). The MHB medium 0.1% (*v*/*v*) Tween 80 and ethanol 1.5% (without EO) was used as a growth control. Broth microdilution was used to determine the MIC of sertraline and EO for these tests. Sertraline was added to each well at concentrations of 128, 64, 32, 16, 8, and 4 μg/mL and EO at concentrations of 9.8, 4.9, 2.4, 1.2, and 0.6 mg/mL. MIC was defined as the lowest concentration of the mixtures at which no visible growth of the bacterial strains could be detected, compared to their growth in the negative control well. MIC values are given in mg/mL and μg/mL for CEO and sertraline, respectively. MIC determinations were performed in triplicate in three independent assays. MIC data of sertraline and CEO were converted into fractional inhibitory concentration (FIC), determined using this formula FIC = (MIC_A_ ^combination^/MIC_A_ ^alone^). MIC values for the EO–sertraline associations were defined as the lowest concentration at which no visible growth of the microbial strains could be detected, compared to their growth in the control well, as described in CLSI and EUCAST document [61].

#### 4.2.2. Microdilution Checkerboard Method

In the combined trials, the checked procedure described by White et al. [62] was used to assess the synergistic action of EO with sertraline. Twelve double-series dilutions of EO were prepared using the same method used to assess the MIC. In our experimental procedure, some modifications to CLSI protocol have been applied. Pure powder of sertraline was dissolved to obtain stock solution concentration of 2 mg; serial dilution was produced in the range of 4–0.5 µg/mL, and for EO, it was produced in the range of 0.5–0.06 mg/mL. A checkerboard testing was carried out in MHB in microdilution plates by using elements from CLSI M27A9. Drug dilutions in two-fold increments were prepared at four-fold levels above the concentration for each compound (drug and EO) tested. Each well contained a combination drug made at 50µL each, in this way forming a 2x concentration of each drug. In the first step of our experiment, the serial dilutions ranged from 40% to 5% for EO and from 25% to 3.12% for sertraline. In the second step, we used a higher percentage of EO (40%, 20%, 10%, 5%) and a lower percentage of sertraline (25%, 12.5%, 6.25%, 3.12%). Microdilution checkerboard method was used to mix all sertraline dilutions with the appropriate concentrations of the EO. In this way, it has been possible to obtain a series of concentrations of concentration combinations of the EO with sertraline. The prepared concentrations represented 40%, 20%, 10%, and 5% of the MIC value for EO and 25%, 12.5%, 6.25%, and 3.12% of the MIC value for sertraline. In our procedure, the combinations of the substances were analyzed by calculating the FIC index (FICI) as follows: FIC of sertraline plus FIC of EO. Generally, FICI value was interpreted as: (i) a synergistic effect when it is ≤0.5; (ii) an additive effect or indifference when it is >0.5 and <1; (iii) an antagonistic effect when it is >1 [54]. Generally, the combination of the two components can be shown graphically in a Cartesian dia. By applying the isobologram, the non-interaction of the two components results in a straight line, whereas the occurrence of an interaction is shown by a concave isobole [57,63]. 

#### 4.2.3. Gas Chromatography/Mass Spectrophotometer Equipment

Gas chromatographic hyphenated with mass spectrometry analysis of CEO was performed on an Agilent 6890 N gas chromatograph equipped with a 5973 N mass spectrometer, provided with an HP-5 MS (5% phenylmethylpolysiloxane, 30 m, 0.25 mm i.d., 0.1 μm film thickness; J & W Scientific, Folsom) capillary column, following the procedure described in our previous paper [35,37,64].

#### 4.2.4. Compound Identification

Pure CEO sample was diluted 1:100 in ethyl acetate and after filtration, 1 μL of this solution was injected into the GC-MS. The GC/MS analyses were carried out in triplicate.

Qualitative analyses were carried out comparing the calculated linear retention indices (LRIs) and similarity index of mass spectra (SI/MS) for the obtained peaks with the analogous data from Adams 4th ed. (Adams, 2007) and NIST 2017 Databases and by comparison with authentic standards available in our laboratory. LRI of each compound was obtained by temperature programming analysis and was calculated in relation to a homologous series of *n*-alkanes (C7–C30) under the same operating conditions, following the Van den Dool and Kratz equation [50] and compared with the arithmetic index (AI) from NIST Chemistry WebBook Database [52] and Adams 4th ed. [51]. Component relative percentages were calculated based on GC peak areas without using correction factors. Chemical determination of structural equation modeling (SEM) was performed using Microsoft Excel.

## 5. Conclusions

The synergistic associations of drugs represent a valid approach in the antimicrobial therapies, since they have provided positive results in recent years. Our previous studies on EOs, based on the synergy with antibiotics or non-antibiotics drugs, demonstrated the effectiveness of these associations. The data reported in this study underline that in vitro CEO possesses a decisive and strong action towards a large panel of bacteria in association with sertraline, a selective serotonin reuptake inhibitor whose antibacterial activity has been successfully confirmed. The sertraline-CEO association resulted in a very strong synergistic mode of action for all Gram-positive and Gram-negative strains tested, as assessed by the FIC indexes whose values were significantly lower than the limit value 0.5. Therefore, the combination of both compounds significantly reduces the amount of sertraline required to inhibit bacterial strains. The results against Gram-negative bacteria as *Klebsiella pneumoniae* and *Pseudomonas aeruginosa* are of particular interest as these bacteria are difficult to treat with commonly employed antibacterial drugs. Further investigation will provide more knowledge of the mechanism underlying the synergism and a more complete understanding of the antimicrobial potential of this association, which could represent a new potential strategy for the repositioning of the antidepressant drug sertraline and may be useful for developing safe drug combinations for the cure of infections caused by these common pathogens.

## Figures and Tables

**Figure 1 antibiotics-11-01617-f001:**
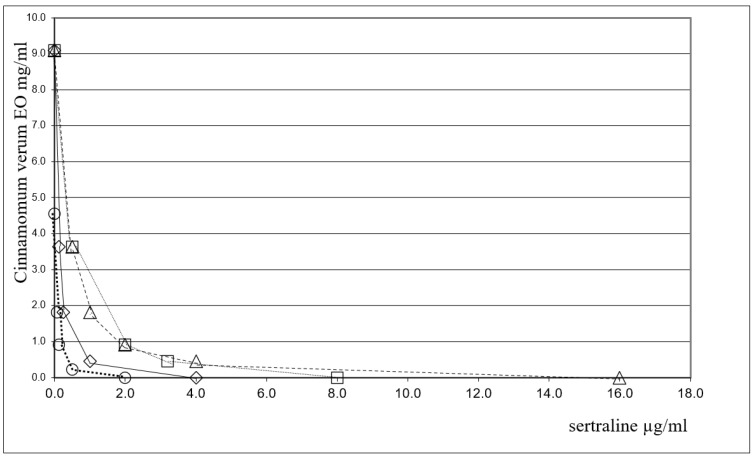
Isobole curves revealing the synergistic effect of sertraline with CEO in inhibiting four bacterial strains. ◇ *P. aeruginosa ATCC 27853,* ☐ *E. faecalis ATCC 29212,* △ *K.pneuomoniae ATCC 13883,* ◯ *B.subtilis ATCC 6633*.

**Figure 2 antibiotics-11-01617-f002:**
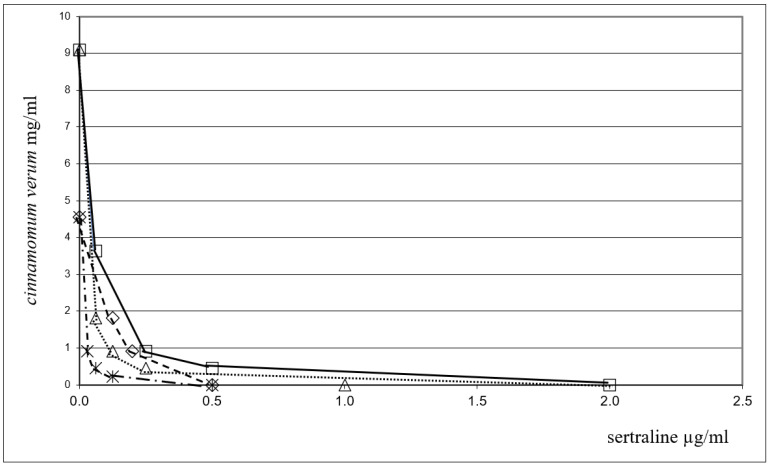
Isobole curves revealing the synergistic effect of sertraline with CEO in inhibiting four bacterial strains. △ *E. coli* ATCC 25922, 🞵 *S. aureus* ATCC 29213, ◇ *B. subtilis* ATCC 6633, ☐ *S. aureus* ATCC 6538P.

**Table 1 antibiotics-11-01617-t001:** Chemical composition of *Cinnamomum verum* EO.

Compound	Peacks Area % ± SEM	Library/ID	SI/MS	LRI	AI
1	0.3 ± 0.04	α-Pinene ^a^	94	934	934
2	0.12 ± 0.06	Camphene	96	950	949
3	0.5 ± 0.06	β-Thujene	91	965	968
4	0.2 ± 0.04	α-Phellandrene ^a^	90	1005	1001
5	0.52 ± 0.04	p-Cymene	95	1021	1021
6	1.47 ± 0.29	Eucalyptol ^a^	98	1023	1023
7	6.78 ± 1.25	Linalool ^a^	97	1095	1098
8	0.16 ± 0.012	o-Anisaldehyde	98	1220	1222
9	72 ± 5.9	(E)-Cinnamaldehyde ^a^	97	1225	1226
10	0.21 ± 0.029	Safrole	97	1285	1287
11	0.3 ± 0.05	α-Cubebene	98	1348	1348
12	6 ± 1.7	Eugenol ^a^	98	1360	1359
13	3.7 ± 0.8	Caryophyllene ^a^	99	1410	1408
14	0.5 ± 0.06	Humulene	95	1452	1452
15	5 ± 1.5	Cinnamyl Acetate ^a^	97	1455	1455
16	0.13 ± 0.023	Eugenol Acetate	96	1525	1524
17	0.24 ± 0.028	Caryophyllene oxide	83	1580	1578
18	0.6 ± 0.08	Benzyl Benzoate	96	1755	1753

^a^: Standard compounds. Linear retention index (LRI) on HP-5MS column was experimentally determined using a homologous series of C7-C30 alkanes standard mixture [50]. Arithmetic index (AI) was taken from Adams 4th Ed. [51] and/or the NIST Database [52]. Similarity index/mass spectrum (SI/MS) was compared with data reported on NIST Database [52] and were determined, as reported by Koo et al. [53] and Wan et al. [54]. Relative percentage values are means of three determinations with a structural equation modeling (SEM) in all cases below 10%.

**Table 2 antibiotics-11-01617-t002:** Antibacterial activity of sertraline (μg/mL), *Cinnamomum verum* EO (mg/mL), and their combination on different bacterial strains.

Strains	MICo ^a^	MICc ^b^	FIC ^c^	FICI ^d^	R% ^e^
*Bacillus subtilis* ATCC 6633					
Sertraline	64.0	4.00	0.06	0.11	94
CEO	1.22	0.06	0.05		95
*Enterococcus faecalis* ATCC 29212					
Sertraline	16.0	0.50	0.03	0.08	97
CEO	1.22	0.06	0.05		95
*Enterococcus faecalis* BN21					
Sertraline	32.0	1.00	0.03	0.13	97
CEO	1.22	0.12	0.10		90
*Enterococcus faecalis* BS					
Sertraline	32.0	1.00	0.03	0.08	97
CEO	1.22	0.06	0.05		95
*Staphylococcus aureus* ATCC 25923					
Sertraline	16.0	0.50	0.03	0.13	97
CEO	1.22	0.12	0.10		90
*Staphylococcus aureus* ATCC 29213					
Sertraline	16.0	0.50	0.03	0.28	97
CEO	1.22	0.31	0.25		75
*Staphylococcus aureus* ATCC 43300					
Sertraline	32.0	1.00	0.03	0.43	97
CEO	1.22	0.49	0.40		60
*Staphylococcus aureus* ATCC 6538p					
Sertraline	64.0	4.00	0.06	0.11	94
CEO	1.22	0.06	0.05		95
*Staphylococcus aureus* BS					
Sertraline	16.0	1.00	0.06	0.26	94
CEO	1.22	0.24	0.20		80
*Acinetobacter baumannii* ATCC 19606					
Sertraline	16.0	0.50	0.03	0.08	97
CEO	1.22	0.06	0.20		80
*Corynebacterium striatum* RM					
Sertraline	32.0	1.00	0.03	0.08	97
CEO	2.44	0.13	0.20		95
*Escherichia coli* ATCC 25922					
Sertraline	16.0	0.50	0.03	0.23	97
CEO	4.88	0.98	0.20		80
*Escherichia coli* ESBL					
Sertraline	32.0	1.00	0.03	0.43	97
CEO	4.88	1.95	0.40		60
*Klebsiella pneumoniae* ATCC 13883					
Sertraline	33.0	1.03	0.13	0.53	97
CEO	4.88	1.95	0.40		60
*Pseudomonas aeruginosa* ATCC 27853					
Sertraline	64.0	8.00	0.06	0.11	87
CEO	4.88	0.24	0.05		95

All the MIC (minimal inhibitory concentration) values for sertraline and CEO are reported in μg/mL and mg/mL, respectively. ^a^ MICo: MIC of single component tested alone; ^b^ MICc: MIC of each component in the association at the most effective inhibition growth; ^c^ FIC^:^ Fractional inhibitory concentrations is determined by the ratio MICc/MICo; ^d^ FICI (fractional inhibitory concentration index): FIC of sertraline + FIC of *C* EO; ^e^ R% represents the percentage reduction in the amount of each associated component, compared to each single component. FIC and FICI are reported as means of three replicates.

## Data Availability

Not applicable.

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
