# Peer review of "Synergistic Action of Cinnamomum verum Essential Oil with Sertraline"

_antibiotics, 2022, doi:10.3390/antibiotics11111617_

Round 1
Reviewer 1 Report
In the manuscript "Synergistic Action of Cinnamomum Verum Essential Oil with Sertraline" giving novel information about combination essential oil with commercial remedy. However, it is missing explanation about main idea for combination these two substances.
I'm not sure I understand table 1. There are given MIC, FIC and FICI for CEO and Sertraline, but not their combination? You used mathematical model for prediction their synergistic effect in figure 1 and 2? Please give more information!
Author Response
Response to Reviewer 1 Comments
In black the reviwer comments and/or questions; in red authors comments and/or answers
Authors: the authors thank the reviewers and the editorial office for their constructive criticism and we hope that the changes introduced in the revised version that we are now submitting may be considered suitable for pubblication in this journal.
Reviewer 1.
R1_Q1_In the manuscript "Synergistic Action of Cinnamomum Verum Essential Oil with Sertraline" giving novel information about combination essential oil with commercial remedy. However, it is missing explanation about main idea for combination these two substances.
R1_A1_Antibacterial drug resistance is a multifactorial phenomenon that is based on several factors including the modification of the target site and the pumping out of the active drug trough specific efflux pumps. Anyway, it is above all the reckless use of antimicrobial agents to causes this phenomenon. Some strategies have been identified to overcome drug resistance, among which the most common approach is to increase the efficacy of existing chemotherapeutic agents by co-administering other drugs that can neutralize acquired resistance by bacterial agents (reference 42 of revised manuscript). On the basis of this idea, recently, our attention has been focused on essential oils whose antimicrobial properties are well described (references 14-19 of revised manuscript) and on some non-antibiotic drugs that may exhibit antimicrobial activity. Generally, in the in vitro assays we observed an increase in the activity of antibacterial agents, reversing the phenotypes of multidrug-resistant bacteria and making them susceptible to the same antibiotics for which they had developed resistance (reference 42 of revised manuscript). One of the most used essential oils is represented by Cinnamomum Verum Essential Oil (CEO), while one of the repositioned drugs for which an off-label effect has been proposed, is sertraline, a serotonin reuptake inhibitor used in the treatment of nervous system disorders, which has demonstrated antimicrobial activity, too.
Moreover, another aspect that can not be underestimated is linked to dyseusia, a side effect that arises with the prolonged use of SSRIs, such as sertraline (1, 2). The use of CEO could still improve this aspect, as it has a sweetish taste due to the cinnamaldehyde and a little bit pungency linked to the presence of tannins, which can cover the strange taste of sertraline, increasing patient compliance (3). We are planning these experiments for our next work in which we are preparing new formulation for in vitro and in vivo administration.
- https://optimistminds.com/zoloft-metallic-taste/
- Schiffman SS. Influence of medications on taste and smell. World J Otorhinolaryngol Head Neck Surg. 2018 Mar 26;4(1):84-91. doi: 10.1016/j.wjorl.2018.02.005. PMID: 30035266; PMCID: PMC6051304.
- Yokomi N, Ito M. Influence of composition upon the variety of tastes in Cinnamomi cortex. J Nat Med. 2009 Jul;63(3):261-6. doi: 10.1007/s11418-009-0326-8. Epub 2009 Mar 18. PMID: 19291358.
R1_Q2_I'm not sure I understand table 1. There are given MIC, FIC and FICI for CEO and Sertraline, but not their combination?
R1_A2_ The main idea from which we started in this project of repositioning of drugs and natural substances, was to verify the synergism of combinations in the antimicrobial activity of already known drugs and essential oils, whose antimicrobial activity has been widely described in the literature. In the table 2 (manuscript revised version) antibacterial activity of sertraline and CEO used as single component (first column) and their combination (second column which represent the MIC of the single component in combination with the other) on different bacterial strains were reported. We determined the synergistic effects of the components in their combination by calculation FIC and FICI parameter following EUCAST definitive document E. Def 2.1 protocol (references 61 of revised manuscript). All the MIC (minimal inhibitory concentration) values for sertraline and CEO are reported in mg/mL and mg/mL, respectively. MICo (first column) is the MIC of single component tested alone; MICc (second column) is MIC of each component in the association at the most effective inhibition growth; FIC Fractional inhibitory Concentrations is determined by the ratio MICc/MICo; FICI (Fractional Inhibitory Concentration Index) is determined by adding FIC of sertraline + FIC of CEO; R% represents the percentage reduction in the amount of each associated component compared to the each single component. The section 2.2 Antibacterial Activity and the table 2 caption are extensively revised for better understand the reported results.
R1_Q3_You used mathematical model for prediction their synergistic effect in figure 1 and 2? Please give more information!
R1_A3_Isobole method or isobologram is a figurative mathematical method as described by Williamson et al. (reference 63 of revised manuscript) that graphically represents the efficacy of two component combination in lower amounts than the single ones. In all our assays the isobole curve appears as concave line, being the effect of association synergistic against the bacterial strains considered (reference 57 of revised manuscript).
Reviewer 2 Report
The authors already developed other studies about essential oils and antibiotics and non-antibiotic drugs, so they have experience in this field. The study is originally and shows the benefic antimicrobial effects of a popular essential oil associated with a antidepressive drug, demonstrating very promising results preliminarly. Furthermore, new studies should be conducted in order to stablished the antimicrobial activicty of sertraline.
Some aspects are pointed below due to improve the quality of the study:
1-Such in the abstract as in the introduction the popular name of Cinnamomum verum can be inserted;
2-The authors can explain better the origin of clinical isolates used for tests;
3-In the initial part of methods the authors cited a reference (number 50) of CG-MS of CEO. So, its seems that the CEO analysis will not be conducted after, however, this analysis was performed in this study. In view to solve this confused situation I suggest to insert the analysis of CEO composition first of antimicrobial activity;
4-Why in the results only two isoboles of each four bacterials were showed and not of all bacterials?
5-It would be important insert a toxicity test using both CEO and sertraline, however, this can be included in next studies;
6-In the discussion I think that it is interesting provide more information about the clinical use of sertraline as antibiotic drug, since it is a central nervous system drug that generat adverse side effects.
Author Response
Response to Reviewer 2 Comments
In black the reviwer comments and/or Questions; in red authors comments and/or Answers
Authors: the authors thank the reviewers and the editorial office for their constructive criticism and we hope that the changes introduced in the revised version that we are now submitting may be considered suitable for pubblication in this journal.
Reviewer 2.
R2_Comment 1_The authors already developed other studies about essential oils and antibiotics and non-antibiotic drugs, so they have experience in this field. The study is originally and shows the benefic antimicrobial effects of a popular essential oil associated with an anti-depressive drug, demonstrating very promising results preliminarily. Furthermore, new studies should be conducted in order to established the antimicrobial activity of sertraline.
Some aspects are pointed below due to improve the quality of the study:
R2_Q1_Such in the abstract as in the introduction the popular name of Cinnamomum verum can be inserted;
R2_A1_The popular and commercial name of different Cinnamomum species is reported as Ceylon Cinnamon or Cinnamon tree and refers to the cinnamon bark commonly used as spices. This popular name has been inserted in the abstract and introduction, taking into account the reviewer suggestion. We carried out our experiments on Cinnamomum verum essential oil obtained by water steam distillation of Cinnamomum verum bark.
R2_Q2_The authors can explain better the origin of clinical isolates used for test;
R2_A2_All the isolates were from patients admitted to the Intensive Care Unit of the Department of Biomedical Sciences and Human Oncology, University of Bari, Italy. The isolation procedures were conducted in the Hygiene Section of the Department, using conventional physiological and morphological methods, API system. This sentence was inserted in section 4.1. Materials.
R2_Q3_In the initial part of methods the authors cited a reference (number 50) of CG-MS of CEO. So, it seems that the CEO analysis will not be conducted after, however, this analysis was performed in this study. In view to solve this confused situation I suggest to insert the analysis of CEO composition first of antimicrobial activity;
R2_A3_The GC-MS analysis was carried out before the biological assay in order to identify the most abundant components which could be the main responsible for antimicrobial activity. In fact, in the literature and in our experience, the antimicrobial activity has already been verified for the components in the whole essential oil. Thus, following the reviewer suggestion, the section 2 Results was revised reporting the paragraph 2.1. Cinnamon verum EO Chemical Composition before the paragraph 2.2. Antibacterial Activity. Moreover, the references were renumbered and revised.
R2_Q4_Why in the results only two isoboles of each four bacterials were showed and not of all bacterials?
R2_A4_The isobologram showed in figure 1 and 2 are representative of the best results in terms of the synergistic effects and are reported for eight out of fifteen bacterial strains. Moreover, in order to obtain the isobolograms, it is necessary to carry out more assays using different doses of the components in their association, resulting in an increase in the economic resources to be committed.
R2_Q5_It would be important insert a toxicity test using both CEO and sertraline, however, this can be included in next studies;
R2_A5_In the literature there are several studies of in vitro cellular toxicity of the essential oil (1). On the other hand, since sertraline is a drug already known and used in clinical practice for a long time, several studies have been published concerning both the in vitro and in vivo toxicity of this molecule (2,3). Anyway, no studies have yet been published regarding the toxicity of sertraline in association with CEO, so we are setting for our next studies the in vitro cell viability assays on cellular models to avoid the cytotoxic effects of our combination.
- Wijesinghe GK, de Oliveira TR, Maia FC, de Feiria SB, Barbosa JP, Joia F, Boni GC, Höfling JF. Efficacy of true cinnamon (Cinnamomum verum) leaf essential oil as a therapeutic alternative for Candida biofilm infections. Iran J Basic Med Sci. 2021 Jun;24(6):787-795. doi: 10.22038/ijbms.2021.53981.12138. PMID: 34630956; PMCID: PMC8487610
- Thomas S. Davies & William M. Klowe (1998) Preclinical Toxicological Evaluation of Sertraline Hydrochloride, Drug and Chemical Toxicology, 21:2, 163-179, DOI: 3109/01480549809011645
- Chen, S., Xuan, J., Couch, L., Iyer, A., Wu, Y., Li, Q. Z., & Guo, L. (2014). Sertraline induces endoplasmic reticulum stress in hepatic cells. Toxicology, 322, 78-88.
R2_Q6_In the discussion I think that it is interesting provide more information about the clinical use of sertraline as antibiotic drug, since it is a central nervous system drug that generates adverse side effects.
R2_A6_ The use and discovery of antibiotics represent a milestone in medicine, but the beginning of drug resistance to antibiotics is becoming a real emergency to deal with. Also, for this reason, research is turning its attention to the approach of repositioning drugs already known for other therapeutic properties and which may show off-label effects such as potential antibiotics. Starting from these bases, we chose to use sertraline in association with the CEO. Sertraline is an antidepressant belonging to the third-generation selective serotonin reuptake inhibitors (SSRIs). As antidepressant, sertraline act increasing serotonine quantity in the synapse space. Instead, in bacteria, its antibacterial activity is due to a probable inhibition of pump efflux. However, further studies have to be done in this area. The in vitro activity of sertraline alone and in combination with other antibiotics against various Gram-positive and Gram-negative bacterial and fungal species has been reported in numerous studies. With these two aspects in mind, in our preliminary in vitro studies, the synergistic antimicrobial activity of sertraline in combination with Cinnamomum Verum essential oil was evaluated. The data in Table 2 show that the antimicrobial activity of the sertraline-CEO combination is achieved at lower doses than the individual components. Although it is not possible to establish the occurrence of adverse effects in this study as it is a preliminary in vitro study, the reduction of the doses of both components of the combination and the different mechanism of action with which the antimicrobial activity of sertraline would develop constitutes an interesting result for the reduction also of the side effects of sertraline itself.
Reviewer 3 Report
The authors provide assessment of combination of plant-derived essential oil mix with repurposed drug. This presents a unique opportunity to blend synthetic and natural compounds for drug-resistant bacterial inhibition. Presumably, this approach is still in its infancy and is not yet ripe for animal, or human testing/trials, despite the approval of both drugs for human contact either via ingestion or injection. The use of the FICI method works to determine drug interactions, but little else.
The manuscript has a generally clear writing style with adequate clarity and descriptiveness. There remains a significant degree of editing needed to correct grammar and some spelling errors. Some specific points are below.
Abstract
synergic or synergistic? Spelling check.
Introduction
P1: EOs exhibit antifungal, antibacterial, and…
P2: While primary components are typically associated with EOs (Suggested revision)
P2: constituents, not costituents (spelling error)
Results
P3: of this combination is reported in Table 1…
P3: isobologram? versus isobole method
Materials and Methods
P7: The culture media used… (Mis-spelling)
P7: 1 x 10^8
4.2.5: Statistical Analysis
Other than the authors’ description of SEM calculation, why do authors indicate as statistical analysis. The authors report no statistical data analysis, and report only an experimental design. FIC and FICIs could be reported as means of replicates to determine how consistent the observed results are. But such presentation is not provided.
Author Response
Response to Reviewer 3 Comments
In black reviewer comments and/or questions; in red authors comments and /or answers
Authors: the authors thank the reviewers and the editorial office for their constructive criticism and we hope that the changes introduced in the revised version that we are now submitting may be considered suitable for pubblication in this journal.
Reviewer 3.
R3_Comment 1_The authors provide assessment of combination of plant-derived essential oil mix with repurposed drug. This presents a unique opportunity to blend synthetic and natural compounds for drug-resistant bacterial inhibition. Presumably, this approach is still in its infancy and is not yet ripe for animal, or human testing/trials, despite the approval of both drugs for human contact either via ingestion or injection. The use of the FICI method works to determine drug interactions, but little else.
The manuscript has a generally clear writing style with adequate clarity and descriptiveness. There remains a significant degree of editing needed to correct grammar and some spelling errors. Some specific points are below.
Abstract
R3_Q1_synergic or synergistic? Spelling check.
R3_A1_The terms synergic and synergistic are both correct and their use depends by the context and “author feeling”. In the scientific literature we found these two terms quite equally reported to indicate a positive effect deriving by the combination of different compunds, methods or activities. In this manuscript as in our previous work we have been used the term “synergistic” instead of “synergic” throughout the manuscript.
Introduction
R3_Q2_P1: EOs exhibit antifungal, antibacterial, and…
R3_A2_P1: “exhibited” has been changed into “exhibit”
R3_Q3_P2: While primary components are typically associated with EOs (Suggested revision)
R3_A3_P2: the sentence “While primary components are typically concerned with EOs biological activities, the role of minor costituents to these activities should be examined.” has been changed into “Although major components are generally responsible for EOs biological activities, the contribution of minor constituents to these activities should be also considered.”
R3_Q4_P2: constituents, not costituents (spelling error)
R3_A4_P2: “costituents” has been corrected with “constituents”
Results
R3_Q5_P3: of this combination is reported in Table 1…
R3_A5_P3: “has been reported” has been changed into “is reported”
R3_Q6_P3: isobologram? versus isobole method
R3_A6_P3 “isobole” has been corrected with “isobologram”
Materials and Methods
R3_Q7_P7: The culture media used… (Mis-spelling)
R3_A7_P7: “colture” has been corrected with “culture”
R3_Q8_P7: 1 x 10^8
R3_A8_P7: 1x108 has been corrected with 1x108
R3_Q9_4.2.5: Statistical Analysis
Other than the authors’ description of SEM calculation, why do authors indicate as statistical analysis. The authors report no statistical data analysis, and report only an experimental design. FIC and FICIs could be reported as means of replicates to determine how consistent the observed results are. But such presentation is not provided.
R3_A9_ the section 4.2.5: Statistical Analysis was deleted. In the section 4 Materials and method and the caption of tables 1 and 2 has been revised reporting the number of replicates.
Round 2
Reviewer 1 Report
Well done. The manuscript is now suitable for publication in the Antibiotics Journal.